# Application of the Water–Energy–Food Nexus Approach to the Climate-Resilient Water Safety Plan of Leh Town, India

**Natalie Páez-Curtidor, Daphne Keilmann-Gondhalekar *** and **Jörg E. Drewes**

Chair of Urban Water Systems Engineering, Department of Civil and Environmental Engineering, Technical University of Munich, Am Coulombwall 3, 85748 Garching, Germany; natalie.paez@tum.de (N.P.-C.); jdrewes@tum.de (J.E.D.)
* Correspondence: d.gondhalekar@tum.de

**Abstract:** Climate-resilient water safety plans (CR-WSPs) have been developed as a risk-based approach to ensure a safe drinking-water supply while addressing the increasing stress on water resources resulting from climate change. Current examples of the application of CR-WSPs show a strong sectoral approach that fails to explore the potential synergies between other climate-sensitive sectors related to water, such as food and energy. This can increase the vulnerability or decrease the overall resilience of urban systems when planning climate change adaptation measures. In this work, the Water–Energy–Food (WEF) Nexus approach was applied in the formulation of a CR-WSP in Leh Town, India, a city with rapid development and population growth located in the Himalayas—one of the most sensitive ecosystems to climate change. The WEF Nexus approach was applied in the system description using a critical infrastructure approach and in the formulation of scenarios for risk management which exploited intersectoral synergies through water reclamation with resource recovery using constructed wetlands. The improvements in WEF security and risk reduction were demonstrated through indicators and risk mapping with geographical information systems (GISs). The methods for integrating the WEF Nexus approach in CR-WSPs provided through this work can serve as a base for a trans-sectoral, resilient approach within risk-based approaches for water security.

**Keywords:** climate-resilient water safety plans; Water–Energy–Food Nexus; water security

## 1. Introduction

More than two billion people worldwide live in areas with high water stress [1]. Population growth, the intensification of agriculture, urbanization, industrialization and the persisting water pollution will further increase the pressure on water resources [1]. As water in sufficient quantity and quality is crucial for life and development, ensuring water supply and sanitation has gained increasing attention in the global political agenda and is embodied in the establishment of many of the Sustainable Development Goals (SDGs) [1]. In particular, cities experience specific challenges regarding water stress due to rapid population growth, growing water demands and an increasing provision of food and energy which accompany economic development [2]. This can lead to an overexploitation of freshwater resources and, in many cases, to an increased dependency on imported water sources or on a single local source, which increases competition in water use between different sectors [2]. The fact that urban water systems depend on global, regional and watershed-scale pressures adds complexity and uncertainty to urban water security [2]. Furthermore, the risk of flooding in cities increases as impervious areas expand, and the rise in wastewater generation together with poor sanitation can adversely impact freshwater quality [2]. In developing cities, the existence of slums and poverty with no proper water and sanitation infrastructure presents an additional challenge [2]. Moreover, the changing patterns in the water cycle that derive from climate change can impact the occurrence of hazardous events, water availability and the water quality of freshwater resources [2,3]. Climate change can also threaten the existing infrastructure for

water provision and wastewater management, which has harsher impacts on communities with less capacity for preparedness due to mismanagement and rapid and unplanned urbanization [3]. The effects of climate change not only affect the urban water environment but also influence all social and ecological systems at various scales, thus conditioning the quality of life and economic competitiveness of cities [4].

Within this context, achieving water security is a recognized priority and has been regarded as the general aim of water management by various authors and institutions [2,5–7]. Although there is no unique definition of water security, its most comprehensive interpretation involves fulfilling all of the different water services for societies and ecosystems while also considering overall welfare, social equity and environmental sustainability [2,3,7]. Thus, water security relates to the pillars of societal security itself [7]. In particular on a city level, urban water security involves applying the water security concept to a "territory of an urban area, municipality or urban agglomeration" [2]. The importance of approaching water security at the city scale lies in the distinct and inherent characteristics of urban areas that influence water stress. These include their high population density, their dependence on regional or global land for the supply of resources—including water, food and energy—and the high density of economic activities occurring within cities, which concentrates water-related risks [2]. In addition, there are particular governance settings at the urban level that usually exist in different municipal departments for different water-related services or indirect water-relevant duties [2].

Water security has been approached in the literature from four main foci, as concluded by Hoekstra et al. [2]: enhancement of economic welfare, increase of social equity, achievement of long-term sustainability or reduction of water-related risks. The risk-based approach implies managing hazards—such as natural disasters and water pollution—and reducing the vulnerability to extreme conditions arising from drivers such as climate change [2]. In urban areas, the exposure to hazards is relatively high derived from the concentration of people and assets, while vulnerability comes from improper adaptation, unpreparedness, low coping capacity and lack of measures for increasing resilience [2]. As risk represents the potential for unwanted events to materialize, risk-based decision making uses evidence of risk to guide individual and societal choices regarding possible courses of action [8]. This approach has been preferred by some authors as they provided a "theoretic, empirical, and operational" basis for water security while allowing the quantification of the effectiveness of investments for addressing water-related risks [8]. The water safety plan (WSP) methodology developed by the World Health Organization (WHO) is an example of risk-based approaches for managing water-related hazards.

The necessity for a complimentary, interdisciplinary approach that recognizes the linkages of water security with food and energy security has been addressed in various publications [2,6,7,9,10], as achieving water security requires sound coordination and collaboration between several stakeholders with competing needs [3]. The Water–Energy–Food (WEF) Nexus approach is an analytical framework and governance approach that acknowledges this relationship, proposes to identify synergies and trade-offs within the sectors and promotes collaboration between them [7,11,12]. This is relevant considering that water, energy and food share critical characteristics such as their increasing global demand, the billions of people that lack access to them, their character as global goods and their dependency on healthy ecosystems [3]. The advantages of the WEF Nexus approach lie in the identification of opportunities to increase resource efficiency and the increase in cooperation and policy coherence between the sectors [12]. At urban level, the importance of this approach is exemplified in the increased dependency of urban areas on external energy, water and food provision which extends the environmental footprint of cities on scales larger than their geographical limits [2]. Furthermore, large-scale water services—such as drinking-water supply and wastewater management—require a high amount of energy for conveyance and treatment [2]. Urban water management interventions such as ecosystem conservation and water reuse can therefore impact energy consumption and carbon emissions significantly [2].

In the following sub-sections, the main characteristics, types and selected outcomes of current WSPs are discussed, as well as the importance of the WEF Nexus in climate adaptation and its possible implementation within the WSP framework.

*1.1. The Water Safety Plan Approach for Water Security*

A WSP is a recognized, well-established risk management approach designed to ensure the safety of the drinking-water supply from catchment to consumers [13]. In a broad sense, WSPs aim to operationalize the framework for safe drinking water (FSDW), also established by the WHO, whose primary goal is to meet health-based targets in drinking water through a preventative approach [14]. To understand the relationship between the hazards and the processes in the drinking-water supply system and to establish preventive actions against hazardous events, WSPs are based on the hazard analysis and critical control points (HACCP) methodology. This involves analyzing hazards, identifying critical control points, establishing critical limits, monitoring, implementing corrective actions, documentation and verification [14]. An additional important pillar of WSPs is the Stockholm Framework, which consists of an integrated risk management approach for controlling infectious water-related diseases [15]. Elements such as risk assessment, community involvement and the multiple-barrier principle have also been implemented in WSPs to prioritize risks and to lower the probability of failure through errors or omissions in the water supply [14]. The WSP methodology has been applied in about 90 countries, both in urban settings and rural communities [16]. Several potential benefits of implementing WSPs include the protection of public health, compliance with regulatory requirements, improved consumer and stakeholders trust, cost-effectiveness and investment planning [16]. Some documented examples of improvements in water supply systems derived from WSP implementation include infrastructure upgrades, decreases in chemical and microbial pollution in water, and a better understanding of hazards and vulnerability against hazardous events, as in the case of Uganda [17] and Bangladesh [18]. An increase in water quality compliance was reported from case studies in Iceland [16,19], and policy development and investment in water quality information systems have been reported in South Africa [16,20]. Cost reduction, improvement in management practices and integration of WSPs within public policies are other reported impacts of their implementation [16,17,21].

In some countries, the WSP methodology has been adopted to the local context and has been integrated into existing water management programs, as in the case of the Pacific Islands, Ethiopia, the Democratic Republic of the Congo and India [16]. In particular, India has focused on the development of water security plans (WSePs) at state level to ensure that every individual in rural areas has enough safe water to meet their daily uses at all times [16,22]. The water management interventions have mainly addressed source management, water system management and water quality monitoring and surveillance, focusing on changing community behaviors through participatory approaches, obtaining external support for training, development of process guidelines, water quality monitoring procedures and infrastructure, infrastructure planning and funding allocation in water provision systems [16].

As the primary focus of WSPs is to meet health-based targets for human health [15], some conceptual modifications of the methodology have arisen for enhancing its scope and addressing broader risks related to the urban water system [14]. Sanitation safety planning (SSP) was proposed by the WHO as a risk management approach for sanitation systems to ensure the achievement of health targets and addresses the risks related to wastewater treatment and reuse in agriculture [23]. This approach has been piloted in cities in India, Peru, the Philippines, Portugal, Uganda and Vietnam, where positive contributions to reducing water pollution and increasing safe water reuse have been observed since its implementation [23]. Water-cycle safety plans (WCSPs) enhance the WSP scope to the urban water cycle for the protection of public health, public safety and the environment, while considering climate change adaptation as the main driver for this integrated approach [15,24]. WCSPs acknowledge the aggravation of current conditions

and the emergence of new hazards arising from the impacts of climate change in the water cycle while pointing out that risk management requires an integrated approach and common work between society, utilities and other stakeholders [15]. WCSPs also acknowledge the dynamic character of the water cycle, which requires considering all of its components and interactions, thus incorporating wastewater management and reuse, as an extension to water supply [15]. This also considers resource efficiency and the minimization of greenhouse gas (GHG) emissions and acknowledges that water quality and quantity need to be addressed [15]. This approach was piloted in Lisbon, Eindhoven, Oslo and Simferopol as part of the UN PREPARED project [15].

More recently, the climate-resilient water safety plan (CR-WSP) methodology has been developed by the WHO to address climate-related impacts on the water supply system in order to strengthen its resilience [24,25]. Although water suppliers already include climate-related events in planning—such as regular heavy rainfall or droughts—many of them have not addressed adaptation to future changes in climate in a systematic manner [25]. Resilience can be defined as the "ability of a system to absorb changes of state variables, driving variables, and parameters, and still persist" [26,27]. In water management, this involves the capacity of maintaining services and supporting livelihoods when facing extreme events while incorporating long-term adaptation [26]. Thus, CR-WSPs highlight the need for adaptation to increasing flood risks and changes in the projected renewable water resources availability, which will intensify the competition for water between different sectors and will affect regional water, energy and food security [25]. The methodology also stresses the need for long-term planning, water demand and competition management, implementation of control measures to ensure water supply in suitable quality and quantity and improving operation and maintenance to ensure its effectiveness [25].

For long-term resilience and supporting adaptation, CR-WSPs recommend the use of the IWRM principles for managing competing needs, increasing water use efficiency and handling trans-boundary water resources [25]. Many of the IWRM principles are linked to institutional and governance arrangements that directly or indirectly help to reduce risks to water supply systems [25]. At urban level, different approaches for implementing IWRM have been proposed, such as the Water Sensitive Urban Design in Brisbane, Melbourne and Sydney (Australia); the Sponge City in several cities in China including Beijing, Xiamen and Shenzhen; or the Sustainable Urban Drainage Systems in cities like Cambridge and Yorkshire (UK) [28–31]. All of these are motivated by the shared vision of shifting water management from the conventional paradigm of centralized, engineering-focused approaches toward more flexible and decentralized systems [30]. In addition, including non-conventional water resources and using innovative solutions for water reclamation with resource recovery are key elements of the application of IWRM in urban contexts [29]. Some case examples of the application of CR-WSPs have been identified in several cities in Africa, Europe, South-East Asia and Western Pacific, most of them focusing on large urban water utilities [24]. The improvement measures proposed within CR-WSPs have included restrictions on land use, using nature-based solutions (NbS) such as reforestation with native tree species to protect the water table, using alternative water sources such as treated wastewater and flood resilience and attenuation measures [15,24,25].

### 1.2. The Role of the Water–Energy–Food (WEF) Nexus in Climate Adaptation

Water, energy and food are one of the most climate-vulnerable sectors [11]. At the same time, energy and food production have an important impact on the global greenhouse gas (GHG) emissions, which makes them influential sectors for climate change mitigation [11]. Since the WEF Nexus approach involves identifying crucial interactions, conflicting demands and potential synergies between WEF sectors, it can be a powerful way to achieve sustainable climate change adaptation [12]. Currently, actions for climate change adaptation have a development-oriented approach that is based on strengthening resilience to climate risks by considering the main causes of vulnerability rather than only responding to its effects [12]. This requires a trans-sectoral approach that is not limited to

infrastructure-based solutions but extends to building response capacities and addressing the socioeconomic and institutional barriers that can also drive vulnerability [12].

According to Rasul and Sharma [12], WEF Nexus approach-based adaptation aims to achieve security in WEF sectors through resource efficiency, intersectoral coordination and policy coherence, which ultimately reduces vulnerability [12]. The main elements of this approach involve minimizing resource waste while maximizing economic efficiency and encouraging sustainable supply, focusing on integrated solutions at multiple scales (local, national and regional) [12]. This approach is complimentary to IWRM in the sense that it expands the sectoral priority on achieving water security to equally important energy and food security [12]. Furthermore, systems thinking is inherent to the concept of resilience, as understanding the complex intersectoral relationships allows one to respond and adapt against risks [26].

From this perspective, possible adaptation measures to manage intersectoral trade-offs can include increasing freshwater use efficiency, substituting freshwater use with alternative water resources such as treated wastewater, increasing resource recovery from wastewater, improving watershed management, increasing the use of renewable energy, improving the productivity of irrigation systems, promoting better crop, soil and nutrient management, rainwater harvesting, shifting consumer behaviors and reducing food waste [11]. Some of these measures overlap with those recommended within the IWRM framework and as such can be implemented within CR-WSPs as supporting programs and risk management procedures.

### 1.3. Research Gaps and Hypotheses

Although the need for an integrated approach is recognized in the CR-WSP framework, current examples of its application show a strong sectoral approach that fails to explore explicitly the potential synergies between other climate-sensitive sectors related to water, such as food and energy. This is exemplified in various country-level guidelines for (CR-)WSPs, where the energy and food sectors are only partially addressed in the system description and hazard identification, as in the example of the guidelines of the Netherlands [32], Ethiopia [33] and Nepal [34]. In the Indian WSP guidelines, the food and energy sectors are addressed as part of the water supply and irrigation infrastructure [22]. However, the interconnections, synergies and trade-offs are, at least explicitly, left out of the analysis and risk management strategies. In general, WSPs are envisioned to be managed by the water supply providers, with external stakeholder collaboration limited to enabling the proper operation of water supply systems. In addition, most examples of improvements derived from the application of WSPs mostly report improvements on the water supply and compliance with water quality parameters [16]. Although this is beneficial for water security, important intersectoral dependencies and potential intersectoral benefits can be overlooked. This can increase the vulnerability or decrease the overall resilience of the system when planning adaptation measures [12].

As the need for a more integrated approach within CR-WSPs is evidenced in the literature, this work aimed to address this gap by exploring the ways in which the Water–Energy–Food (WEF) Nexus approach can be applied to the formulation of a CR-WSP in Leh Town, India—a city with rapid development and population growth located in the Himalayas—one of the most sensitive ecosystems to climate change [35]. This work hypothesized that incorporating the WEF Nexus approach into the CR-WSP of Leh Town can contribute to the management of water-related risks while addressing synergies between food and energy for an enhanced WEF security in a climate change context. The town is located in a remote region with limited natural, financial and human resources and already faces several challenges regarding WEF security [12]. With an increasing population, a rapidly expanding tourism industry and a rising economy, Leh Town faces challenges for coping with an increasing WEF demand while also experiencing groundwater overextraction and pollution from on-site sanitation facilities. Although water management at river-basin scale is one of the paradigms of the implementation of IWRM [36], the WEF

Nexus in CR-WSPs was evaluated at urban scale in this work. As WEF sectors are most commonly managed within urban administrative boundaries, limiting the boundaries of water security at city level enables a more direct operationalization of water security, as pointed out in [37,38]. The following section provides an overview of the town and its WEF sectors.

## 2. Case Study Site: Leh Town (Ladakh, India)

Leh Town is the capital of Ladakh, a region located in the Trans-Himalayas which was declared as a Union Territory of India in 2019 (Figure 1) [39,40]. It is the administrative and cultural center of Ladakh and is one of the towns with the most rapid growth in the country [40]. With an average altitude of 3500 m above sea level, the town lies in a remote, semi-arid area characterized by an average annual precipitation of 61 mm [41]. Precipitation and cloudbursts occur mostly during the monsoon period [35]. Leh has a cold desert climate, with an average temperature that can reach −28 °C in winter up to 35 °C in summer [35].

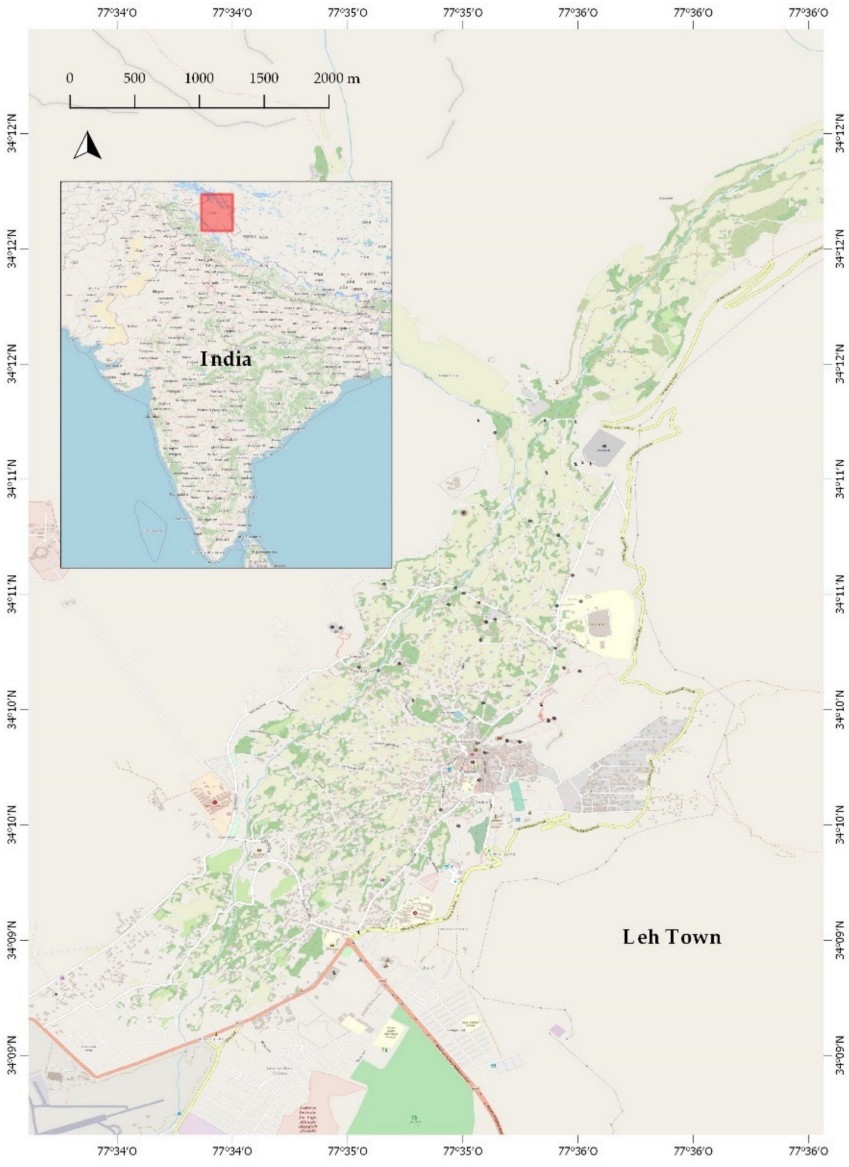

**Figure 1.** Geographical location of Leh Town, Ladakh, India.

The Leh valley lies within the Ladakh Plutonic Complex [42,43]. From a geological perspective, the study area is composed of granitoid, fluvio-glacial deposits and quaternary sedimentary rocks [44]. A hard-rock fractured aquifer fed from precipitation and glacial- and snowmelt constitute the main freshwater resource of the town [44]. Due to the presence of high lineament density, high drainage density, lower slopes and the porosity of the surface and subsurface in the center of Leh valley, the aquifer below Leh Town has a high groundwater potential and serves as a groundwater accumulation zone [44]. The groundwater flows from north to south following the terrain gradient and discharging into the Indus River [42,45].

Leh Town used to have a traditional agricultural society [39]. However, tourism has expanded rapidly since Ladakh was opened for visitors in the 1970s [39,44]. As most of the tourists visit the region during the summer months (April through October), the population of Leh Town has a high seasonal fluctuation. In 2018, it was estimated that about 35,807 people inhabited the town during the whole year, while the floating population reached around 65,927 people, including tourists and seasonal migrant workers [44]. The resident population in Leh Town has expanded exponentially since the 1990s, mostly driven by the economic expansion created by tourism [39]. It is estimated that the fixed population in the city will reach 43,000 people by 2021 and 55,000 people by 2031, showing a vast difference compared with the 9897 inhabitants of the town in 1991 [39]. Ladakh is also an important strategic region with an increasing regional presence of the Indian Army. Daily, about 20,000 army personnel are stationed in Leh [39].

The local government, the Ladakh Autonomous Hill Development Council (LAHDC), issued, in 2020, a vision document for 2030 [39] acknowledging the need to understand and manage the challenges that the region has experienced since its opening to tourism. Through a participatory multi-stakeholder process, several planning strategies were proposed to address issues in mobility, water, sanitation, solid waste management, climate change, heritage conservation, social infrastructure, spatial planning, urban design and governance, based on a shared vision of wellbeing [39]. This demonstrates a high level of awareness of the local government regarding the threats and opportunities that Leh Town is facing confronted with rapid urbanization, economic change from agriculture to the tourism sector, shifts in consumption patterns and climate change.

## 2.1. Consequences of Climate Change in Leh Town

Mountainous regions have high sensitivity to the impacts of climate change and are exceptionally vulnerable to the amplified changes in temperature and precipitation patterns resulting from it [35,46,47]. The climate of the Himalayan region is influenced by the climate at global, regional and local levels and has a serious impact on the climatology, hydrology and ecology of India [35]. A statistical analysis of the climate of Leh performed in 2018 showed an overall increase in temperature and precipitation since the 1990s and reported some indication of an increasing number of days with high precipitation in the region over the years [35]. These results envision a further changing climate with a higher probability of unexpected events, as well as impacts on ecology, vegetation, wildlife, hydrology, cryosphere and agriculture in Leh [35].

Despite the typically dry climate, severe flash flooding events have been reported in the region, as in the case of the 2010 flashfloods [48] which resulted in more than a hundred deaths and several damages in Leh Town [39]. The damages included the destruction of water-related infrastructure such as the man-made ponds (*Zings*) that have an important role in water provision for agriculture [40]. Another flashflood in 2015 destroyed irrigation canals and fertile agricultural land while wiping out several crops, livestock and households [49]. Water shortages, loss of local power generation and damage in distribution networks were additional consequences of this event [49].

Although the occurrence of cloudbursts has not been extensively monitored in the region, it has been estimated that about six additional flashfloods have happened in the Leh district since 2005 [48], and there are signs of an increased frequency of these events over the

last decade [49]. The expansion of tourism, the rapid urbanization and the establishment of infrastructure in flood-prone areas further increase the vulnerability of the region against this hazard [49]. In addition, the observed increasing trend of precipitation and temperature in the region further increases the vulnerability to extreme events and cloudbursts which can have a critical impact in Leh [35].

The availability of groundwater and surface water depends on the glaciers located north of Leh Town, namely the Phutse and the Nagste glaciers [50]. A study from 2017 that evaluated the change of high altitude glaciers in the Trans-Himalaya reported that the area of the Phutse glacier decreased by 19% between 1969 and 2016 [50]. The area of the Nagste glacier decreased by 14% during the same time period. Changes in glacier dynamics directly affect the agriculture, livelihood and sustainability of meltwater-dependent communities [50], such as Leh Town.

### 2.2. The Water Sector in Leh Town

The increased number of hotels and guesthouses derived from the expansion of tourism has consequently increased the water demand and the pressure on the freshwater resources of the region, particularly on groundwater [28]. As most of the touristic facilities provide flushing toilets and showers, both the water consumption and wastewater production have increased [36]. There is indeed a difference in water consumption between different social groups in the town: while tourist and local residents consume around 100 and 75 l per capita per day (lpcd), migrant workers receive only 35 lpcd [35]. The estimated domestic water demand in summer is about 5 million l per day (MLD) [51].

Traditionally, 90% of the water supply of Leh Town depended on surface water streams fed from snowmelt, and the remaining 10% of the supply depended on springs [51]. Currently, surface water streams are no longer fit for drinking-water use due to direct discharges of wastewater from commercial establishments and poor solid waste management [40]. In addition, several springs have dried out, which citizens have linked with the increasing groundwater extraction [40]. In fact, 90% of the currently provided domestic water comes from groundwater resources, mostly from the shallow aquifer beneath the town [51].

The public water supply is managed by the Public Health Engineering (PHE) department, which supplies drinking water to the population through household taps, public stand posts and water takers for some parts of the city [51]. About 1.5 MLD of water is pumped from the riverbank of the Indus River to several reservoirs at different altitudes of the town, with an overall capacity of about 7 MLD [51]. Inside Leh Town, about 1.3 MLD of drinking water is extracted from tube wells and borewells [51]. An additional 0.8 MLD of water is taken from several springs in the northern part of the town [51]. The water infrastructure includes five tube wells on the riverbank of the Indus River (3 in operation), 16 storage reservoirs (12 in operation) and a piped water distribution system of about 93 km. Several handpumps are also used to pump water from the shallow aquifer in Leh Town from a maximum depth of 10 m [52]. There are eight public water tankers that supply water to about 2000 households where tap water connections are not available [51].

The tap water provided by the PHE department is only available for 2–3 h per day and covers only 70% of the household water demand in the areas of the town with the largest concentration of guesthouses and hotels [40]. The latest water audit report revealed that water losses through the distribution system add up to 27% due to leakages [40]. The same report also mentions a domestic water supply deficit ranging from 1.67 to 5.09 MLD in summer. Several other issues have been identified in the water supply system, including lack of or non-operating connections, poor or no pressure, irregular supply timings, inconsistent water tanker supply, poor water quality and no volumetric metering [40].

The perceived deficiencies in the public water supply, together with the changes in water consumption behavior driven by tourism, population increase and economic growth, have incentivized the installation of several private borewells for extracting water from the

shallow aquifer of the city [51]. It is estimated that 4000 borewells are currently installed in Leh Town without any regulation on their pumping rates [40], which constitutes additional stress on the groundwater resources.

There are ongoing plans to decommission public stand posts and installing tap water connections for all households, as part of the Jal Jeevan Mission [40,51]. However, there are concerns regarding its feasibility considering the winter conditions in Leh Town, where extreme temperatures lead to frequent pipe freezing and bursting [51]. The water audit also revealed that people are concerned about the increase in water consumption that a 24 × 7 system would yield considering the falling groundwater levels and the glacier reduction from climate change [40].

Adding to the concerns in the household water supply, there is no centralized or unified wastewater management strategy in Leh Town [51]. For the most part, raw sewage is currently disposed of using soak pits or leaking septic tanks without any treatment [41]. These on-site sanitation systems have replaced traditional dry sanitation practices—the Ladakhi dry toilets—from which fecal matter was collected and composed to be used as organic fertilizer [40,52]. Although the Ladakhi dry toilets are still used by the majority of the local population, most of the tourists visiting Leh use flush toilets instead [52]. The groundwater is therefore prone to contamination from these on-site sanitation systems, a fact that has been studied and proven in some studies [40–42,53].

Groundwater pollution is a significant concern, as more than 90% of the drinking water provided to the town depends on these resources [51]. Many private borewells are located at less than 30 m from wastewater discharge points, which puts them at risk of contamination by domestic wastewater [42]. Chloride, nitrate and total dissolved solids (TDS) are well-known indicators of groundwater contamination linked to anthropogenic activities [42]. A study from Schwaller [42] during the summer of 2017 revealed an increased, statistically significant concentration of these parameters in 9 of the 11 wards of Leh Town with respect to the groundwater in the north of Leh, where the population density is lower. In the same study, the dominant source of these pollutants was identified as improper wastewater management systems [42]. In addition, high concentrations of dissolved organic carbon (DOC) were identified as downgradient of pit latrines, soak pits or leaking septic tanks. The increased concentration of these pollution parameters coincides with the increased density of households, guesthouses and hotels in these areas [42]. High pumping rates from borewells create depression cones around the wells, which increases the risk of wastewater seepage from on-site sanitation facilities into the deep groundwater [42]. This is shown in the microbial contamination with *E. coli* identified in most of the deep groundwater samples from the same study [42]. The prevalence of waterborne diseases is a concern in Leh Town, as 64 cases were reported in 2018 [39].

To address these issues, the PHE department started the construction of a centralized sewerage system and a sewage treatment plant in the south of Leh Town [39]. The system is expected to cover 60% of the town with about 62 km of pipelines, with a wastewater treatment capacity of 3 MLD using an aerated activated sludge system [39,54]. Although this would mitigate groundwater contamination, concerns have been reported regarding an overestimation of the population growth for the treatment design, the lack of local experience for operation and maintenance of the system and the location of some parts of the sewage pipes planned to run alongside freshwater provision pipes or areas with high groundwater table [54]. The high volumes of water required for flushing the pipe system and the high energy consumption of the treatment process are additional concerns of the effectiveness and sustainability of the system [54]. The currently constructed sewage lines already show problems in the connections, causing wastewater discharges into open spaces [39].

The Municipal Committee of Leh Town commissioned, in 2017, a fecal sludge treatment plant in the southern part of the town [39], which currently collects and treats about 12,000 L of wastewater per day from septic tanks around the town. This volume is insufficient for the daily produced wastewater, which adds up to 4.2 MLD [39]. All of the

above-mentioned facts point out that it is imperative to seal existing septic tanks, stop the construction of soak pits and open wastewater discharges and evaluate alternatives for decentralized wastewater management systems that complement the planned centralized system.

The first approach for developing a WSP for Leh Town was proposed in 2020 through a collaboration between the Ladakh Ecological Development and Environmental Group (LEDeG), a local non-governmental organization, and members of the Nexus@TUM group (including the authors) at the Technical University of Munich (TUM). This work was conducted within the "Livable Leh" project, whose main goal consisted of capacity building for climate change adaptation [55]. The WSP points out that the greatest perceived risks to the water supply of Leh Town are related to climate change (including changes in glacial melt, changes in precipitation patterns and changes in water availability related to these events), natural disasters (flashfloods), industries (tourism) and water management practices (unregulated groundwater extraction with private borewells and use of soak pits for sanitation).

### 2.3. The Energy Sector in Leh Town

Ladakh has a broad renewable energy potential. Hydropower is an important energy source in the region with an installed capacity of 113 MW with plants such as the Nimoo Bazgo (45 MW) and Chutak (44 MW) in the Indus River [56]. In early 2021, the construction of eight additional hydropower projects in the Kargil and Leh districts with a total capacity of 144 MW was cleared by the Indian government [56].

Leh also receives an important amount of daily solar radiation (about 5.5 kWh/m$^2$), which is higher than the average in India [57,58]. The potential for solar energy generation is yet to be exploited and is gaining attention through the Ladakh Renewable Energy Development Agency created by the LAHDC [59]. Within the Ladakh Renewable Energy Initiative Project, several solar power plants have been constructed in the Leh District with a total capacity of about 1.7 MW [59]. Currently, tender documents have been prepared for the construction of photovoltaic water pumps in cooperation with communities and farmers [59]. Several other projects for solar power harnessing have been implemented as solar greenhouses, water heating systems and solar cookers [59].

As hydropower plants are difficult to operate in winter due to reduced flow stream, thermal energy from diesel-operated plants is used as a backup [60]. Although diesel use is expected to decrease after the connection of Ladakh to the Indian grid in 2019, several sectors still use these fossil-fuel-based plants. For instance, every pump house in the water extraction and conveyance scheme from the riverbanks of the Indus River has diesel backup generators [60]. Some pump houses operate on these diesel generators as they are not connected to the main electricity net [60]. In total, it has been estimated that the current water supply in Leh emits about 658 t $CO_{2\text{-eq}}$ per year [58]. If fossil fuels are also used for pumping through the private borewells, the GHG emissions are even higher. The planned centralized approach for wastewater treatment, which has a high energy consumption for the aeration process, has estimated emissions of almost 2 kg $CO_{2\text{-eq}}$ per cubic meter, almost twice as high as the emissions from alternative decentralized wastewater treatment approaches with constructed wetlands [58].

### 2.4. The Food Sector in Leh Town

Leh Town used to be self-sufficient regarding food production [60]. Small-scale agriculture was a common practice for most families until recent times [39], where land-use patterns have shifted from agriculture to tourism-related activities [52]. Currently, over 90% of the hotels and guesthouses are located in the agricultural wards of Leh Town, and several agricultural lands have been either replaced by buildings or become barren [52].

The water management practices for agriculture have also changed significantly. Historically, irrigation water was distributed and managed by *Churpons*, local leaders elected by the *Goba* (headman of the village) [40]. Water used to be allocated through a

rotational system designed for an equitable surface water distribution depending on the crops grown by each farmer [40]. Leh Town depended mostly on surface water available from snowmelt, and about 18 springs, 3 major streams and 13 major ponds were available for the community, for both household consumption and agriculture [40]. However, many of the previously available springs have dried up, and various ponds have been destroyed by natural disasters or left unused due to lack of management [40]. Although there are still 17 *Churpons* in Leh Town, this practice has decreased over the years, contributing to the degradation of agricultural land and the increase of unmanaged bare land [40]. Currently, farmers are increasingly depending on groundwater to irrigate their farms [39,40].

Despite the decrease in agricultural land, the variety of crops grown in the town has expanded from five types of vegetables in the 1960s to more than 23 types for commercial scale in Ladakh [39]. Broccoli, cabbage, cauliflower, peas, apricots, apples, melons, herbs and medicinal plants are produced in Leh [39]. Pastoral farming is also prevalent in the region [60]. While there is a recognized potential for larger food production within Leh, challenges regarding the long and cold winters and the increasing water scarcity call for implementing newer practices for crop growing and harvesting [39]. This is gaining relevance as the dependency on food imports increases in Leh, which raises several concerns for food security related to the risks of disruptions in the supply chain caused by natural disasters (e.g., landslides), harsh winter weather conditions and global-scale threats such as the COVID-19 pandemic [60]. The aim for increasing food self-sufficiency and becoming an example of a town with water-sensitive, high-altitude urban farming is one of the visions of the town [39], which can contribute to keeping a healthy diet and being a source of local additional income [60].

## 3. Materials and Methods

The WEF Nexus approach was implemented into the CR-WSP of Leh Town in two main parts. On the one hand, a critical infrastructure analysis of the WEF sectors was conducted, intended to compliment the system description of the CR-WSP. On the other hand, two scenarios for risk management that exploit intersectoral WEF synergies were modeled for Leh Town. The scenarios included the implementation of water reclamation with resource recovery using constructed wetlands, using GIS for risk visualization and scenario analysis.

### 3.1. WEF Nexus Analysis of Leh Town with an Urban Critical Infrastructure Approach

The critical infrastructure (CI) approach proposed by [4] was used for identifying the interlinkages within WEF sectors in Leh Town. For this goal, the main characteristics of the CIs regarding the WEF sectors in Leh Town were analyzed, as well as their relation to climate change risks. Then, the interrelationships between WEF sectors were summarized in an interdependency matrix that categorizes the effects of one CI failure in another CI into three categories: low, medium and high. Lastly, the potential implementation of this information in the WSP of Leh was analyzed. The analysis was carried out with support from the Ladakh Ecological Development and Environmental Group (LEDeG) during several meetings and literature review.

### 3.2. Mapping the Risk of Wastewater Seepage from On-Site Sanitation Systems

According to the risk assessment performed for the WSP of Leh Town, an important hazard for its water resources is the seepage of untreated wastewater into the groundwater. This hazard has been linked with the use of inadequate on-site sanitation facilities (i.e., soak pits, leaking septic tanks and disposal of wastewater in streams and canals). However, the geographical distribution of this risk is not evident in the current WSP.

Geographic information systems (GISs) can be a valuable tool for implementing various aspects of WSPs, as they can be used to visualize possible risks and water-related information [61]. A semi-quantitative approach has been used previously for creating an overview hazard and risk assessment [61,62], monitoring of control measures and incident

management [62]. In this work, GIS was used for providing a visual representation of the risk of wastewater seepage from on-site sanitation systems in Leh Town, with the objective of identifying the places where groundwater pollution may be occurring at a higher rate. The semi-quantitative risk mapping was also used for visualizing the impact of the implementation of decentralized wastewater reclamation systems—a WEF Nexus solution—using constructed wetlands for groups of hotels/guesthouses and households. The maps were drawn using the OpenStreetMap base map of Leh Town [63] and the software QGIS 2.18.21 (Las Palmas de Gran Canaria, Spain) [64].

The CR-WSP methodology defines the risk associated with a hazard or hazardous event as the "combination of the likelihood of the hazard or hazardous event occurring (over some time frame) and the severity of the consequences of the hazard or hazardous event if and when it occurs" [25]. The parameters for the risk components were estimated based on literature review and spatial analysis. The maps were designed to represent the conditions of the wastewater seepage risk during the summer months, as this represents the peak in population—and wastewater production—of the town. For risk mapping, the severity was characterized as a function of the pollution load produced on-site that can seep into the ground water. Therefore, this variable depended on the volume of wastewater produced and its quality. The latter depends on the type of wastewater treatment applied. Likelihood was characterized as a function of the number of on-site sanitation facilities with a lateral distance lower than 30 m from borewells, as this increases the probability of wastewater seepage into groundwater.

The parameters were calculated following the methods and assumptions listed in Table 1. For the mapping, an existing spatial database from a survey previously conducted by Gondhalekar et al. in 2015 [52] was used. The database included data with the location and attributes of the PHE tube wells, the Indus River borewells, service reservoirs, hand-pumps, water conveyance pipes, guesthouses/hotels and households. It also included the future location of the planned centralized sewage treatment plant and canals, which will service about 40% of the town [51].

After calculating the parameters, each dataset was reclassified into five groups for obtaining a simplified ranking of the variables. As the data distribution of the wastewater volume and the number of on-site sanitation facilities within 30 m of wells was skewed, the Jenks natural breaks classification method [65] was used. This is a more suitable clustering method in this case as it aims to reduce the variance within each class while maximizing the variance between the classes, and thus the values are better arranged and classified for mapping [65].

**Table 1.** Methods and assumptions for the calculation of parameters for the wastewater seepage risk.

| Risk Component | Parameter | Method | Assumptions |
|---|---|---|---|
| Severity | Wastewater volume (m³/d) | For hotels: multiplied the number of beds by the water consumption of tourists. For households: multiplied the number of people per household by the water consumption of locals or migrants, depending on the ward. | Differences in water consumption depending on population sectors (tourists: 100 lpcd, locals: 75 lpcd and seasonal workers: 30 lpcd) [51]. Wastewater is 80% of water consumption. Full occupation of hotels and guesthouses. Wards of Skampari and Nimoling are mostly inhabited by migrant workers [51]. Four people inhabit each household. |
| | Wastewater quality (mg/L or CFU/100 mL) | Calculated depending on the sanitation system in place, considering the following parameters: biochemical oxygen demand (BOD), total suspended solids (TSS), total nitrogen (TN) and total coliforms (TC). These parameters were chosen as the most relevant for designing decentralized sanitation systems in Leh Town [54,66–68]. | Wastewater generated from households, guesthouses and hotels has the same characteristics as described in [66,69]. The removal efficiency from septic tanks was calculated based on [70]. The removal efficiency of the planned centralized wastewater treatment system was calculated based on [71,72] (the latter for total coliforms only). The removal efficiency for the decentralized wastewater treatment facilities using constructed wetlands was based on [66]. |

**Table 1.** *Cont.*

| Risk Component | Parameter | Method | Assumptions |
|---|---|---|---|
| Likelihood | Number of on-site sanitation facilities within 30 m of wells | Created a buffer of 30 m around each handpump, private borewell and PHE tube well. The number of on-site sanitation facilities within this range was counted. | A distance lower than 30 m between wells and on-site sanitation facilities increases likelihood of groundwater contamination [73]. In total, 10% of the households and 60% of the guesthouses in the database were randomly selected and assumed to have a borewell, following previous survey results [52]. |

For the case of the wastewater quality parameters, the clustering was performed by dividing into five groups the range of pollutant concentration values within those of untreated wastewater (maximum value) and those of treated wastewater intended for Class A water reuse in irrigation as established by the European Union [74] (minimum value). These guidelines were considered suitable for the clustering, as the proposed WEF Nexus solution is envisioned to be applied for water reclamation in agriculture and these guidelines are more stringent than those for treated wastewater for disposal into watercourses. The results of the clustering are presented in Table 2. The 1-to-5 ranking for each variable was used for creating a weighted heatmap. The water quality measurements performed by Schwaller in 2017 [42] were mapped over the risk maps for the base case for comparison.

**Table 2.** Parameter clustering and ranking for semi-quantitative spatial visualization of the wastewater seepage risk.

| Rank | Wastewater Production (m³/d) | Wastewater Quality (mg/L or CFU/100 mL) | Number of Wells <30 m from on-Site Sanitation |
|---|---|---|---|
| 1 | ≤0.64 | BOD: ≤10 mg/L<br>TSS: ≤10 mg/L<br>TN: ≤15 mg/L<br>TC: $\leq 1.0 \times 10^1$ CFU/100 mL | ≤1 |
| 2 | (0.64–1.68] | BOD: (10–70] mg/L<br>TSS: (10–132.5] mg/L<br>TN: (15–22.5] mg/L<br>TC: $(1.0 \times 10^1 – 2.5 \times 10^6]$ CFU/100 mL | (1–3] |
| 3 | (1.68–2.88] | BOD: (70–130] mg/L<br>TSS: (132.5–255] mg/L<br>TN: (22.5–30] mg/L<br>TC: $(2.5 \times 10^6 – 2.5 \times 10^6]$ CFU/100 mL | (3–5] |
| 4 | (2.88– 5.12] | BOD: (130–190] mg/L<br>TSS: (255–377.5] mg/L<br>TN: (30–37.5] mg/L<br>TC: $(1.0 \times 10^1 – 2.5 \times 10^6]$ CFU/100 mL | (5–9] |
| 5 | >5.12 | BOD: >190 mg/L<br>TSS: >377.5 mg/L<br>TN: >37.5 mg/L<br>TC: $>1.0 \times 10^7$ CFU/100 mL | >9 |

### 3.3. Implementing Scenarios for Constructed Wetland WEF Nexus Solutions for the CR-WSP

A major option for implementing a trans-sectoral approach that addresses the risks in the water sector in Leh Town is water reuse with resource recovery. Water reuse, or reclamation, refers to the treatment of wastewater with a defined process and water quality criteria appropriate for an intended final purpose [75]. Water reuse reduces the supply of freshwater resources, which implies savings in water treatment and conveyance, preserves the current freshwater supply and reduces the discharges of wastewater [76]. In addition, water reuse is considered as a reliable water supply that is independent of seasonal droughts and weather patterns, and it is able to cope with peaks of water demand [76]. Due to these advantages, water reuse as an alternative water source is included in current national and international water management strategies and its use is considered as an indicator of increased urban water security [37,77]. As water reuse

widens the portfolio of available water resources, it adds flexibility against water stress, which in turn increases resiliency [26].

For Leh Town, several alternatives for the implementation of decentralized water reclamation systems using constructed wetlands (CWs) have been evaluated [54,66–68]. CWs are a type of nature-based treatment systems (NBTS) that mimic natural ecosystem principles for water treatment. The efficiency of these systems depends mainly on the complex interaction between soil, water, vegetation and the atmosphere, which leads to achieving several treatment goals through mechanical, chemical and biological processes [78,79]. CWs can achieve an appropriate treatment efficiency with relatively low cost of earthwork, piping and construction [80]. As such, they have several advantages: design simplicity, cost-effectiveness, high potential for nutrient recovery and water reuse [80]. An additional advantage is that the main processes in these systems do not rely on external energy resources [81], which increases the sustainability of the treatment as the demand for fossil fuels can be reduced.

In particular, subsurface flow CWs (SSFCWs) have been identified as suitable systems for the climatic conditions in Leh [66]. This type of CWs consists of a basin with an impermeable barrier at the bottom and the sides to prevent seepage of wastewater and groundwater contamination [81]. The basin is filled with porous media (such as sand, soil or gravel) and local plants [81]. The wastewater flows under the surface of the porous media, which minimizes the health risk of wildlife and people from exposure to untreated wastewater [81]. As the water is not in contact with the atmosphere, these systems are appropriate for a place like Leh, where the cold weather can hinder the treatment efficiency of CWs. Suitable process configurations with constructed wetlands envisioned for water reclamation with resource recovery have already been proposed for Leh Town in [66].

The implementation of CWs requires enough available area for the treatment processes. In addition, as the effluent from the CWs is envisioned to be used for agriculture, enough surrounding agricultural land for the treated wastewater is advantageous. Thus, feasible locations for constructing these decentralized water reclamation systems were identified using the OpenStreetMap [63] vector feature classes obtained through the QuickMapServices QGIS plugin. The land-cover layer "landuse = farmland" was downloaded to represent the agricultural land. The land-cover layers "landuse = meadow", "natural = wood" and "natural = scrub" were downloaded for identifying the bare land available for the treatment systems.

The possible available bare land was subsequently identified by defining a setback distance between the treatment system locations and all households, guesthouses, wells, rivers and reservoirs according to the guidelines defined in [82]. From this filtered bare land layer, possible bare land areas that fulfilled the following criteria were selected: they are within 50 m from each guesthouse/hotel and household, downstream from these wastewater point sources (using available digital elevation model data and contour lines) and do not cross paved roads or rivers. Then, the agricultural land nearby the identified bare land for CWs was identified to evaluate the possibility of local irrigation with reclaimed water.

The use of CWs for decentralized water reclamation in agriculture at town scale was exemplified in two scenarios, which could both compliment the planned centralized sewage collection and treatment system:

- Scenario 1: Here, a part of the town is connected to a centralized sanitation system (households and guesthouses within 20 m of the sewage canals are assumed to be serviced, as in [68]), and CWs are built for clusters of guesthouses and households. The CWs for this case were designed for treating the wastewater generated during summer (population peak) using the treatment train presented in Figure 2 [66]. This scenario was the recommended treatment scheme in terms of treatment efficiency, cost and appropriateness for weather and population fluctuations in Leh [66]:
- Scenario 2: Here, the conditions of scenario 1 are complemented by the implementation of CWs for household clusters. The CWs for this case were designed for year-round

operation and do not require the installation of septic tanks in each household. The treatment train for this case is presented in Figure 3 [66].

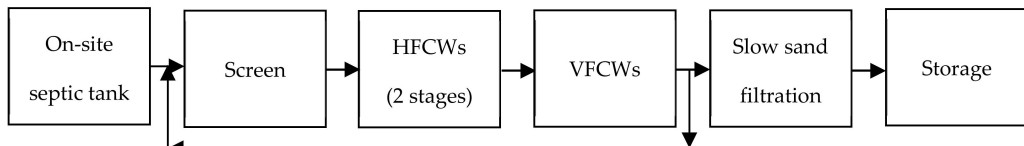

**Figure 2.** Treatment train proposed for clusters of guesthouses and households. Reprinted with permission from ref. [66]. Copyright 2020, Zahra Alipour Tesieh.

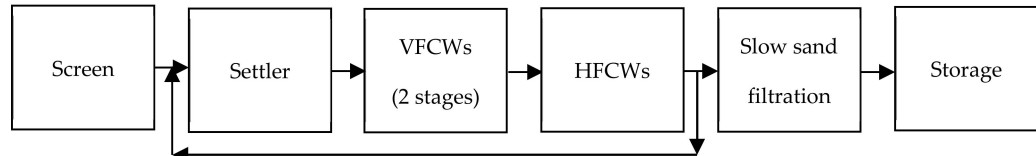

**Figure 3.** Treatment train proposed for clusters of guesthouses and households. Reprinted with permission from ref. [66]. Copyright 2020, Zahra Alipour Tesieh.

The requirements and potential benefits in terms of the WEF Nexus for both scenarios were estimated using previously calculated values reported in [66] for the chosen treatment systems. A detailed study of the irrigation requirements for Leh Town and the potential of using the mentioned CW systems for water reclamation [83] was also used to estimate the potential benefits for agriculture in terms of potential irrigable area and crop production. The most important parameters are listed in Table 3.

**Table 3.** Parameters for evaluation of benefits of WEF Nexus solution in Leh.

| Parameter | Scenario 1 | Scenario 2 |
|---|---|---|
| Total area requirement per cubic meter of wastewater | 33 m$^2$ | 28 m$^2$ |
| Estimated capital costs [1] per cubic meter of wastewater | 2093 EUR | 1207 EUR |
| Sludge production for potential fertilizer use | 1.4 L/PE | |
| Average irrigation requirement in Leh Town per day | 50 m$^3$/ha | |

[1] Excluding land and piping costs.

## 4. Results

### 4.1. WEF Nexus Analysis of Leh Town with an Urban Critical Infrastructures Approach

The description of Leh Town presented in Section 2 and the further literature review of the study site allow identifying important interdependencies within WEF sectors in Leh Town, which are summarized in the current sectoral interdependencies matrix presented in Figure 4:

- Water–Energy: From interviews with the PHE department in 2017, the energy requirement for the four-stage Indus River pumping system requires a total of approximately 3250 kWh, with a daily diesel consumption of about 700 L [42,58,84]. The water tankers, which travel daily an average of 70 km, use about 17.5 L of diesel per day [58]. The increasing installation of private borewells for domestic consumption, agriculture and tourism could increase the fossil fuel consumption of Leh Town. It is estimated that the future expansion of the water supply system envisioned by the PHE will result in a daily diesel consumption of 3600 L [84]. The planned centralized sewage system will have a high water consumption requirement for flushing the pipe system and high energy consumption related to the treatment process [54,58]. Surface water availability is crucial for the hydropower projects present in the city that are increasingly replacing fossil fuel use [60].
- Water–Food: Due to the low available rainfall, agriculture in Leh Town depends on irrigation, which used to be exclusively obtained from glacier-fed surface water

streams and springs but is now increasingly dependent on groundwater abstraction. Bonanno [84] estimated a daily average irrigation requirement during summer of about 275 L/s, which adds up to an average of about 2 MLD. The two main irrigation methods in Leh Town are flood and furrow irrigation [84], both well known for their high water consumption particularly in places with high evapotranspiration rates such as Leh. The food imports to Leh within subsidized food provision programs add up to about 10,000 t of rice, 6000 t of wheat and 19 t of sugar yearly [60]. In addition, about 73% of food grains come from outside of the region, and the food and vegetable import dependency rates are 85% and 67%, respectively [85]. This contributes to the virtual water footprint of Leh Town.

- Energy–Food: Although most of the agriculture in Leh is practiced in small-scale farming systems with prevailing human and livestock labor [86] with low energy consumption, the increasing dependence on groundwater pumping for irrigation contributes to enhancing the energy consumption of the food system. In addition, as importing food to Leh requires shipping by trucks across the Himalayas for at least 430 km from other cities [85], food imports have an impact on fossil fuel consumption and GHG emissions. Furthermore, food production within the Leh District required about 533 MT of fertilizers and 980 l of pesticides in 2018 [85]. Although these are lower than the Indian average consumption values, these influence the energy (and water) consumption along the life cycle of food production in the town.

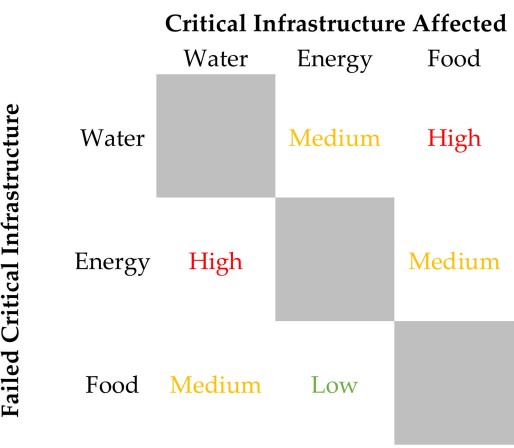

**Figure 4.** Interdependency matrix of the WEF sectors in Leh Town.

The effects of climate change on the WEF sectors of Leh Town are also noticeable from the literature review and the work with the LEDeG. The dependency of agriculture on snowmelt streams, the reliance on groundwater for drinking-water supply, the important role of hydropower in the energy supply and the effects of increasing temperatures in agriculture pose additional challenges to the resilience of the WEF sectors in the town. The effects of climate change can increase the interdependency between the water–energy and water–food sectors, which, added to the pressures derived from rapid urbanization and the expansion of tourism, threaten the resilience and sustainability of the WEF system in Leh. In addition, the use of fossil fuels for groundwater pumping, drinking-water transport and food imports further impacts climate change mitigation actions in Leh.

### 4.2. Mapping the Risk of Wastewater Seepage from On-Site Sanitation Systems

Figure 5a shows the application of the spatial model for visualizing the risk of seepage from on-site sanitation systems in Leh Town, as well as the location of important elements of the drinking-water infrastructure. As observed, the areas with the highest risk are those with a higher population density and a higher concentration of hotels and guesthouses. An important accumulation of high severity values is observed in the touristic wards of the town, where most of the hotels and guesthouses are located. The high severity of

wastewater seepage in these areas is explained by the increased wastewater generation derived from the higher water consumption of tourists. A larger amount of high likelihood values is observed in the middle and southern part of the town, where an increased number of borewells that are within 30 m of on-site sanitation systems is present.

Figure 5b to Figure 5d show a comparison of the risk map with the groundwater quality measurements of nitrate, TDS and *E. coli* performed in 2017 by Schwaller [42]. As observed, there are about nine nitrate measurements with a concentration greater than 50 mg/L located in the center and southern part of the town that coincide with the high-risk areas mapped by the model. Although most of these points correspond to shallow groundwater measurements (from samples in handpumps with about 10 m depth), some measurements from deep groundwater (from samples in private borewells and PHE tube wells with an average depth of 33 m) also show an increased nitrate concentration. In general, increased chemical pollution values are observed in areas with a greater estimated risk of wastewater seepage. This trend is observed for nitrate, TDS and chloride measurements (the latter not shown), which indicate groundwater contamination from anthropogenic on-site sanitation systems [42].

Several values of microbial contamination also coincide with high-risk areas mapped by the model. The greatest values of pollution with *E. coli* are observed in the deep groundwater measurements from private borewells in tourist lodgings in the center of the town. As pointed out in [42], deep groundwater contamination can be explained by high pumping rates which increase the ratio of influence around the wells, making them more prone to pollution from leaking sanitation facilities. The high density of wastewater generation combined with the large number of sanitation facilities nearby private borewells explains this increased microbial pollution, which coincides with the risk map. Although the risk-mapping model is not intended to be a sophisticated water quality model, there is agreement between the water quality measurements and the high-risk areas.

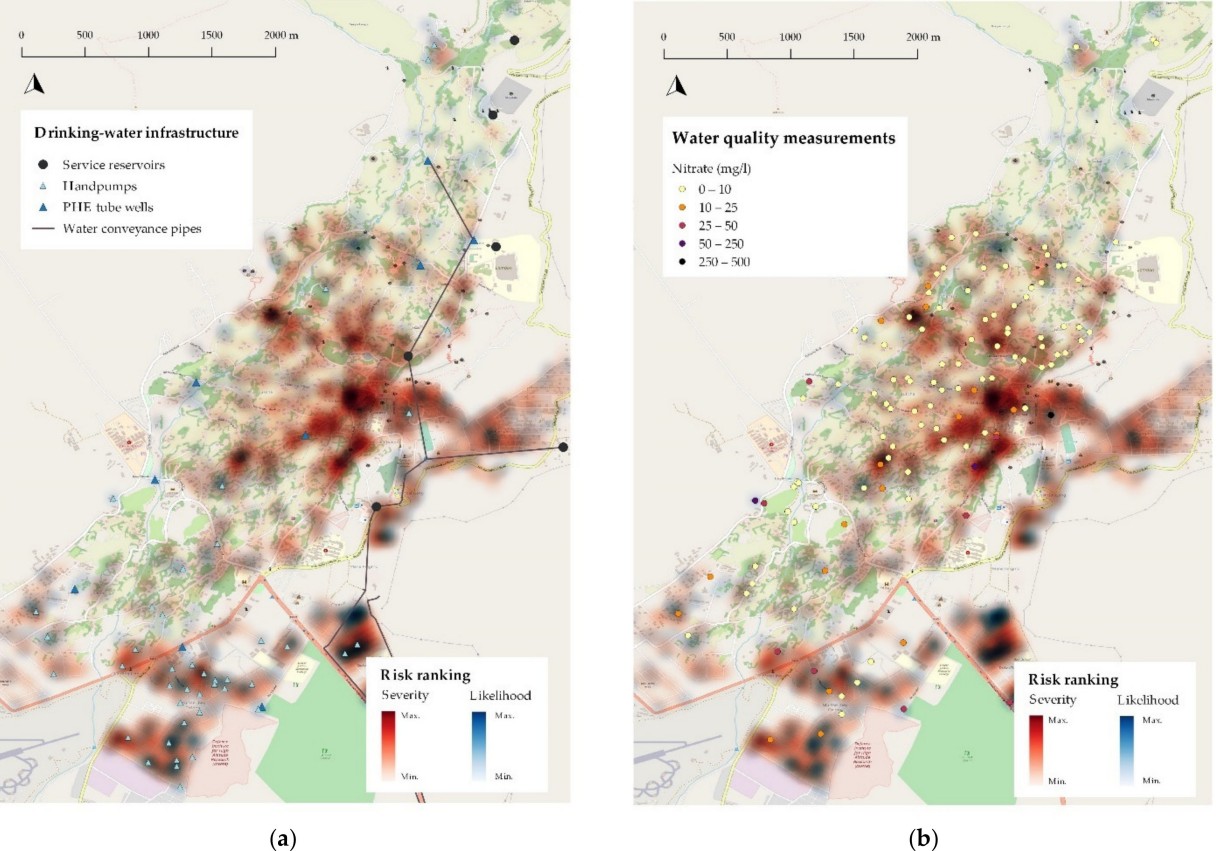

(**a**)  (**b**)

**Figure 5.** *Cont.*

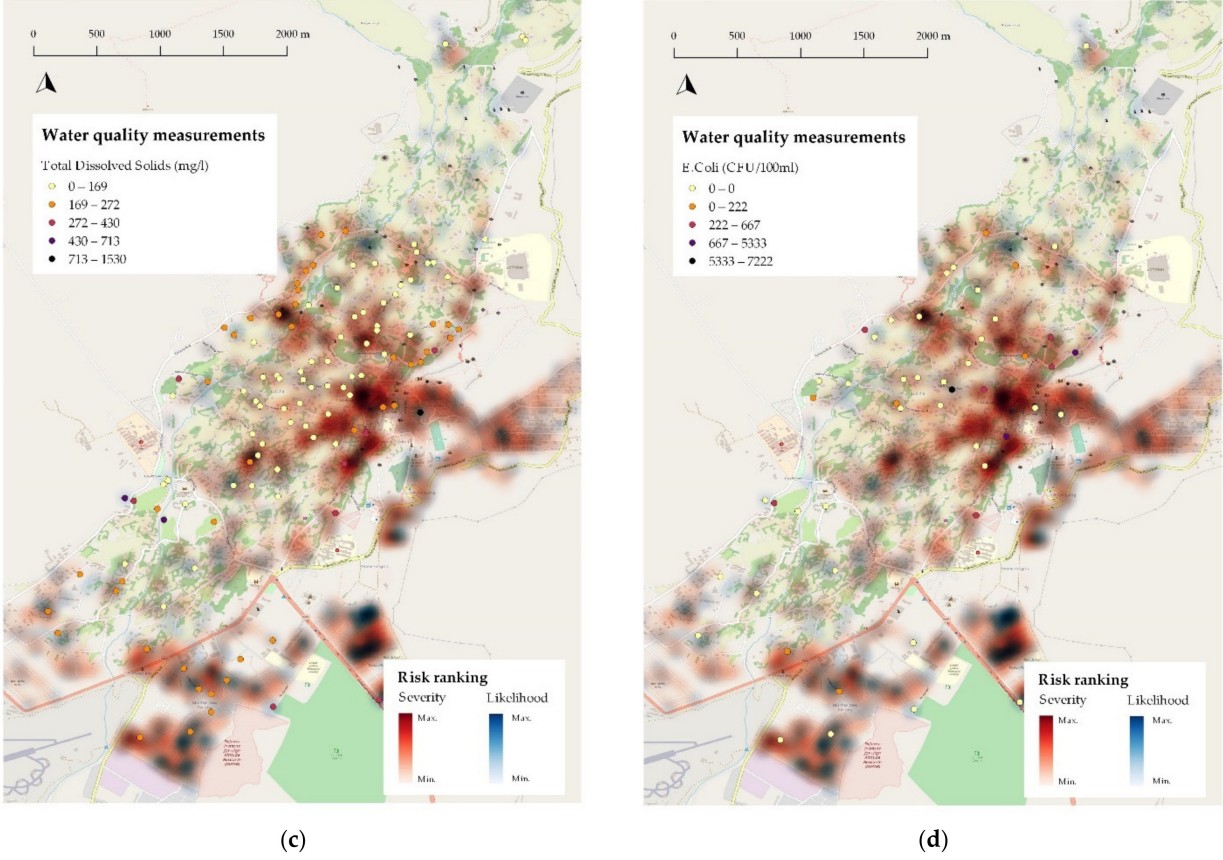

<div align="center">(<b>c</b>)                                                        (<b>d</b>)</div>

**Figure 5.** (**a**) Visual representation of the current risk of wastewater seepage into the groundwater of Leh Town. (**b**) Overlay of nitrate measurements [39]. (**c**) Overlay of TDS measurements [39]. (**d**) Overlay of *E. coli* measurements [39]. Water quality data reprinted with permission from ref. [39]. Copyright 2017, Christoph Schwaller.

### 4.3. Implementing Scenarios for Constructed Wetland as WEF Nexus Solutions for the CR-WSP

Figure 6 shows the WEF Nexus opportunity maps for the two proposed scenarios. In both cases, the spatial analysis showed that there is enough available bare and agricultural land that enables local water reclamation with CWs at neighborhood level. In the case of scenario 1 (Figure 6a), the possible bare land adds up to around 34 ha, with about 74 ha of nearby available agricultural land for local food production. Considering a distance of maximum 50 m between each building, the decentralized water reclamation systems could be designed for treating the wastewater from clusters composed of 2 to 13 hotels, guesthouses and households.

Figure 6b shows the change in the spatial distribution of the risk of wastewater seepage after implementing the sewage management strategies described for scenario 1. The map shows a shift in the risk distribution toward the western side of the town, where most of the population will not be connected to the centralized sewage treatment plant. The drastic reduction of the risk density values in the connected parts of Leh is explained by the reduction of the on-site disposed wastewater volume (severity reduction) and by the transport of wastewater through the centralized canals which would reduce the number of wells within 30 m of sewage discharge points (likelihood reduction). Similarly, the risk density for the areas surrounding the possible CWs also decreases locally, as the wastewater treated on-site would comply with Class A quality for water reuse in irrigation as established by the EU regulations [74] (severity reduction).

The WEF Nexus opportunity map for scenario 2, shown in Figure 6c, shows several additional possibilities for CWs envisioned for wastewater reclamation for household clusters with year-round operation, mostly in the southern and northern parts of the town outside the main touristic areas. Although the volume of wastewater generation is

significantly lower than that from tourist lodgings, there is an enhanced potential for water reclamation. For this scenario, the total possible bare land adds up to around 70 ha, with about 114 ha of nearby available agricultural land for local food production. Considering a distance of maximum 50 m between each building, the decentralized water reclamation systems could be designed for treating the wastewater from clusters composed of two to nine households.

The changes in the spatial distribution of the risk after implementing scenario 2 are shown in Figure 6d. In a similar manner as in scenario 1, the values of risk density around the possible areas for CWs are reduced with the improvement of local sanitation practices. However, the additional risk reduction derived from implementing scenario 2 appears to be marginal compared to scenario 1. This can be explained by that the amount of wastewater produced per household is on average significantly lower than that produced by guesthouses and hotels. Furthermore, the additional CWs are located in areas with a lower population density, thus resulting in a lower representation in the severity and likelihood heatmaps. Nevertheless, the heatmap of the risk shows that an important part of Leh Town still has the risk of seepage of wastewater into groundwater, even after the implementation of both centralized and decentralized sanitation within the conditions described for this scenario. This is explained by the remaining population that is not covered by any of the proposed sanitation systems and is assumed to have leaking septic tanks or soak pits in place. For these cases, an alternative solution must be considered, either concerning the sealing of existing septic tanks or soak pits with proper servicing of the FSTP or connection to the centralized sewage system.

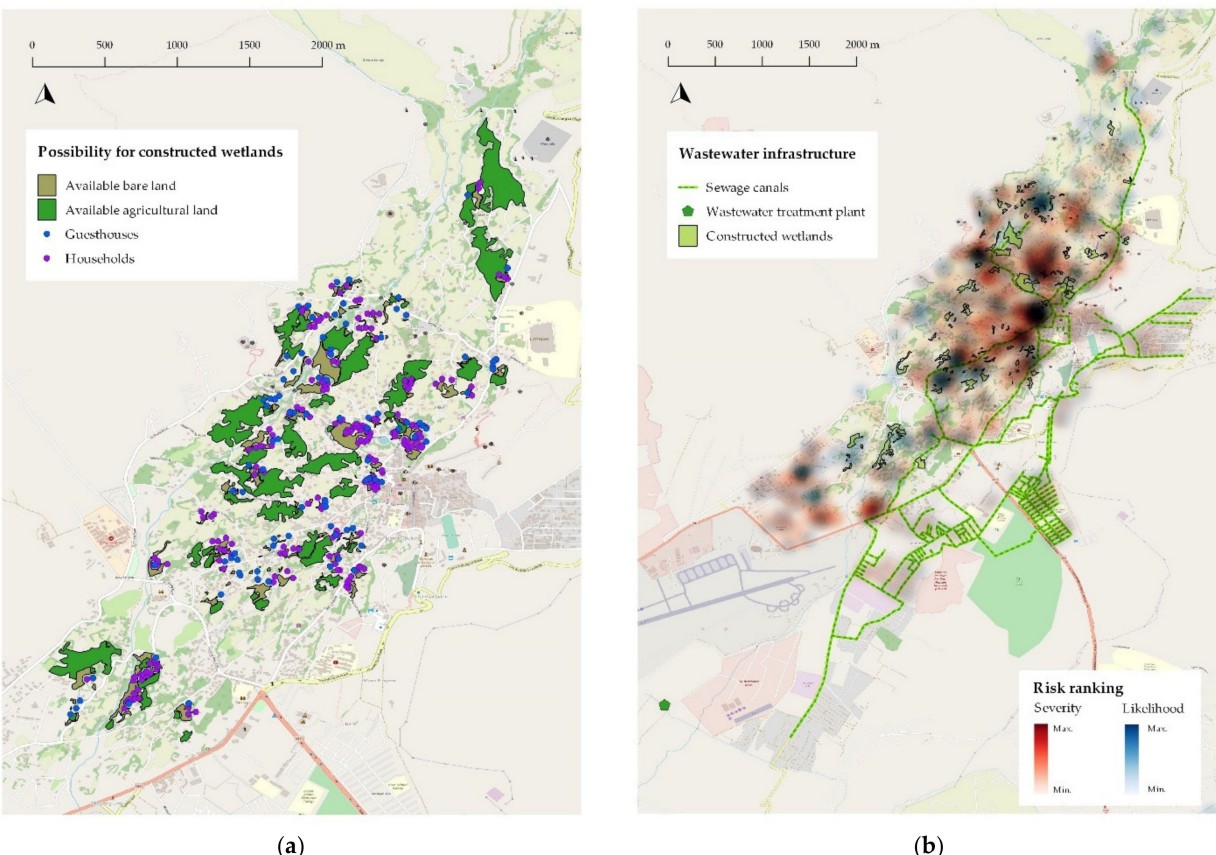

(a)  (b)

**Figure 6.** *Cont.*

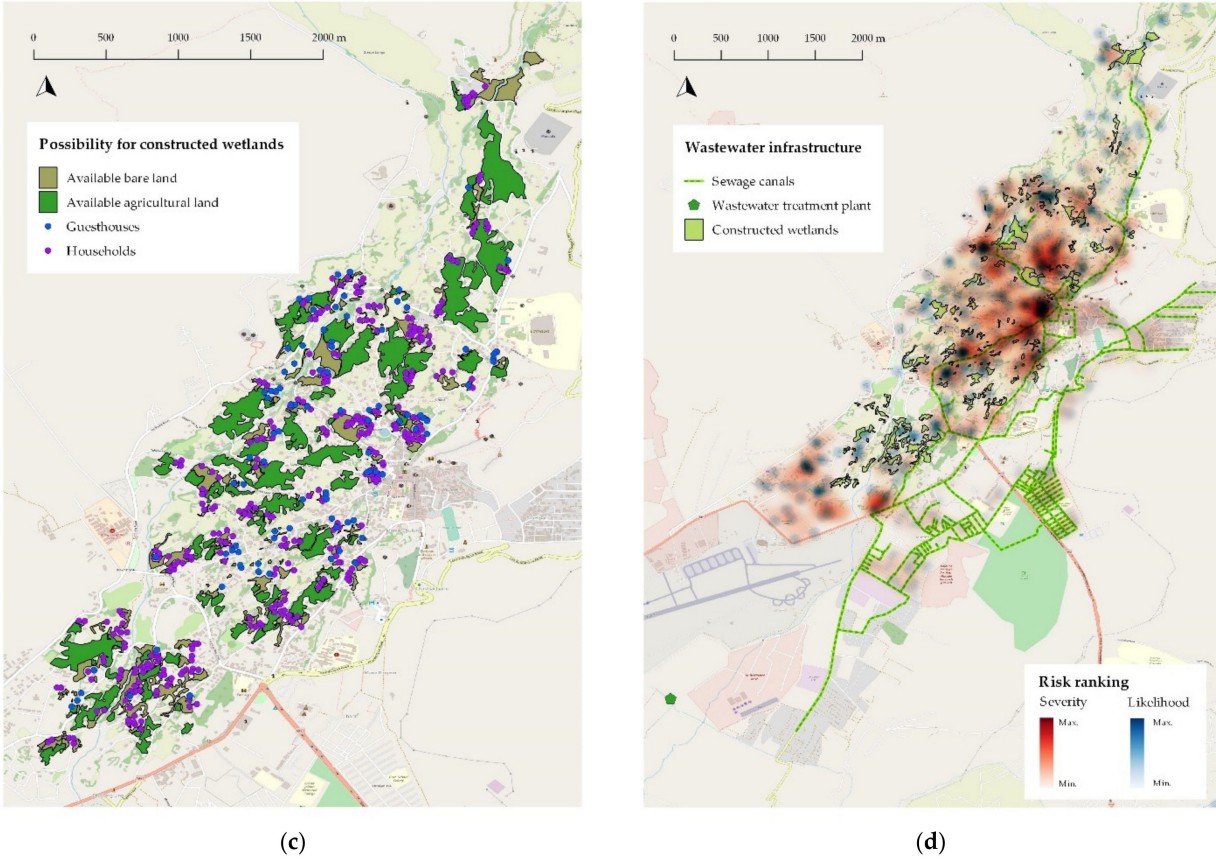

(**c**)　　　　　　　　　　　　　　　(**d**)

**Figure 6.** On the left: WEF Nexus opportunity maps with location of possible available bare land and agricultural land for decentralized water reclamation with constructed wetlands: (**a**) scenario 1; (**c**) scenario 2. On the right: risk maps after implementing the proposed scenarios: (**b**) scenario 1. (**d**) scenario 2.

As observed in Table 4, there can be important additional advantages of implementing both scenarios in terms of the WEF Nexus, in addition to the reduction of the risk of groundwater contamination from wastewater seepage. The volume of reclaimed water available for irrigation could be as high as 275 m$^3$/d in the whole town if scenario 2 were implemented. This volume could cover the irrigation demand for about 55 ha of agricultural land, and up to 155 t/y of fertilizer from composted sludge could be recovered for food production. The potential irrigable area represents about 8% of the estimated planted area in the Leh District, which currently amounts to 678 ha [85]. The chemical fertilizer use in the Leh District in 2018 was about 106 t [85], meaning that the fertilizer recovery from scenario 2 could potentially replace the entirety of the demand in the district.

**Table 4.** Summary of characteristics and benefits from both scenarios.

| Parameter | Scenario 1 | Scenario 2 |
|---|---|---|
| Number of GHs served | 92 | 92 |
| Number of HHs served | 253 | 424 |
| Total area required for CWs | 7675 m$^2$ | 10,155 m$^2$ |
| Estimated capital costs | 573,416 EUR | 622,962 EUR |
| Amount of reclaimed water | 234 m$^3$/d | 275 m$^3$/d |
| Total irrigable area | 47 ha | 55 ha |
| Sludge production (fertilizer) | 95 t/y | 115 t/y |

Table 5 shows the potential local crops that could be irrigated with the reclaimed water from both scenarios. The table also shows the maximum percentage of the crop production in the Leh District that could be covered with the implementation of the scenarios, based

on data from [85]. It can be observed that an important part of the production of onions, cabbage and cauliflower could be produced with reclaimed water from the proposed CWs. In the case of carrots and radish, the available reclaimed water would amply overpass the current production, which suggests a great potential for these crops given their high yields. Although the percentage of local production of potatoes and peas is significantly lower, the potential produced amounts of these crops can be important at neighborhood or household scale. In optimal operation, the proposed system meets the EU guidelines for Class A use, which enables irrigation of all food crops independently of the irrigation method. However, if the system works in its lower efficiency, it complies with a lower water quality class, which would require additional control measures to minimize the microbial risk for water reuse in agriculture. These additional control measures include the use of drip irrigation, crop washing, peeling and cooking, as recommended by the WHO [87].

**Table 5.** Potential of crop production with reclaimed water with the proposed scenarios.

| Crop | Scenario 1 (t/y) | Scenario 2 (t/y) | % of Production in Leh (Scenario 2) |
|---|---|---|---|
| Onion | 84 | 99 | 25% |
| Cabbage | 123 | 145 | 36% |
| Cauliflower | 142 | 167 | 79% |
| Carrot | 316 | 371 | 153% |
| Radish | 169 | 199 | 209% |
| Potato | 102 | 120 | 1% |
| Pea | 33 | 39 | 3% |

## 5. Discussion

Current examples of CR-WSPs are strongly focused on the water sector, which can overlook the potential synergies and trade-offs with other climate-sensitive sectors, such as food and energy. This can lessen the resilience or increase the vulnerability of adaptation measures conceived within CR-WSPs. This work explored some ways for implementing the WEF Nexus approach into the CR-WSP methodology using Leh Town as a case study. The WEF Nexus approach was applied in the CR-WSP of Leh Town in two main parts: First, by including a critical infrastructure analysis of the WEF sectors to identify the current interdependencies and their potential intensification derived from climate change. Second, by proposing scenarios for risk management that exploit intersectoral synergies that contribute to enhanced WEF security in the town. For this aim, the use of a GIS was a central method for risk visualization and assessment of risk management scenarios.

The WEF analysis helped to explicitly highlight the current interdependencies and their implications facing the effects of a changing climate. This allowed identifying relevant additional risks that were overlooked in the current risk assessment of the water supply system of Leh Town. For instance, the increasing dependence of agriculture on groundwater extraction, the persisting dependence on fossil fuels for groundwater pumping and the growing demand for imported food and fertilizers were not considered in the initial risk analysis. This enables an integrated overview of the challenges and opportunities that can be exploited within CR-WSPs. Highlighting these additional risks is enabling a more integral overview of the challenges and opportunities for water resource management in the town, which aids in identifying possible risk management strategies that foster intersectoral collaboration, higher resource use efficiency and the evaluation of integrated solutions at different scales. In this sense, using the WEF Nexus approach in the system description of CR-WSPs can allow a deeper awareness of the interconnections in urban WEF systems, which empowers the evaluation of strategies that enhances resilience and reduces vulnerability to changes in resource availability at the city level.

The involvement of WEF sectors in the system description also aided the selection of suitable alternatives for risk management that considered the intersectoral dependencies and extended the benefits to other sectors. Most current examples of improvements derived

from the application of WSPs report only improvements on the water supply infrastructure and compliance with water quality parameters [16]. In this case, the groundwater contamination derived from the use of improper on-site sanitation facilities was approached with decentralized water reclamation systems designed for treating the wastewater up to suitable quality standards for irrigation and to enable sludge recovery as fertilizer. This way, the proposed scenarios addressed the most critical risk to the water supply of Leh Town and showed positive impacts on the WEF security through the local production of reclaimed water, fertilizer and crops. The proposed scenarios aimed to increase the coverage of proper sanitation, reduce the groundwater demand for irrigation and increase water source diversity in Leh Town. As pointed out in [37,77], these are indicators of increased urban water security. The positive impacts on food security can be appraised through the partial substitution of food imports with locally grown products. The lower energy consumption from the replacement of groundwater pumping and the relatively lower energy requirement of the constructed wetlands versus a fully centralized sanitation approach have beneficial effects on the energy security of the town. There are added benefits of the proposed scenarios in terms of generation of income from the commercialization of locally produced crops. According to the crop market values in Leh reported in [83], the total crop production of scenario 2 represents between EUR 19,000 and 286,000 yearly, depending on the crop. This additional income could have important impacts in equity and economic wellbeing for the town. These results serve as an example of the benefits of integrating a WEF Nexus approach within CR-WSPs, as it can enable the implementation of strategies that contribute to fulfilling different resource management and socioeconomic targets in urban scenarios. This approach also encourages trans-sectoral collaboration for managing competing needs, which strengthens the local capacity for adapting to changes in resource availability at city level.

In this sense, addressing the WEF Nexus enabled the consideration of solutions that enhance intersectoral collaboration instead of more conventional approaches for water management. For example, some other measures that could also mitigate the risk of wastewater seepage would not necessarily increase the overall WEF security of the town. For instance, a fully centralized sanitation system for the whole town with the envisioned wastewater treatment plant would have a high energy requirement due to the activated sludge process, with estimated GHG emissions between 4600 and 7500 t $CO_2$-eq per year (assuming a population of 90,000 to 120,000 PE) [58]. In contrast, a mixed approach where a part of the town is connected to the centralized sewage system and constructed wetlands with gravity-driven sewers are implemented in the unconnected areas (such as those proposed in the present study) has estimated GHG emissions between 13 and 53 t $CO_2$-eq per year for these areas [58]. The centralized sanitation system would also require more water for flushing the system, which can incentivize faster shifts from dry and traditional sanitation practices to flush toilets. This is relevant in a context where the future availability of freshwater has uncertainties related to the changes in the water cycle that climate change already has shown in the Himalayas. This exemplifies the importance of considering water management strategies that exploit synergies with the energy and food sectors, as they foster policy coherence for climate change adaptation and mitigation at urban level. The extent to which the benefits of using a WEF Nexus approach apply for other particular locations depends on the local resource conditions, present risks, interacting sectors and competing interests. Thus, the specific application of the WEF Nexus approach in a framework for managing urban water-related risks needs to be evaluated at a local level with a strong stakeholder engagement approach.

The spatial analysis with GISs proved to be a critical tool for risk visualization and for the evaluation of alternatives for decentralized water reclamation with resource recovery. For this purpose, the most important variables that influence the seepage of wastewater into the groundwater of Leh Town were identified, categorized and operationalized using literature, survey data and feedback from local stakeholders. The agreement between previously conducted water quality measurements and the high-risk areas identified by the model

suggests its suitability to visualize the spatial distribution of this risk, which can aid further decision making and prioritization for localized interventions. The semi-quantitative risk mapping applied in this work needs to be reviewed and modified according to expert opinions and new information on the hazard. Nevertheless, this approach could be applied to other urban systems and hazards with parameters based on risk assessment and stakeholder participation. Although it is possible to use more complex modeling of the spatial distribution of risks, such as modeling of scenarios recreating climate change impacts, the present simplified approach was a useful element for decision making within the WSP framework. These results concur with previous studies, where the importance of risk maps for aiding WSPs was also demonstrated [17,61,62]. For instance, in the WSP of Kampala, Uganda, GISs were used for describing the vulnerability of drinking-water provision pipes combining environmental, infrastructural and socio-demographic data, which was reported to be crucial in defining the priority for implementation of control measures and monitoring [17]. In some cities in Germany, using a semi-quantitative approach for risk mapping allowed an approximate overview of risks to the groundwater resources depending on land use and flooding events [61]. This work shows an additional example of the benefits of using GISs as an important part of the WSP process.

The spatial analysis also allowed the identification of possible areas for implementing the constructed wetlands and to visualize their impact on risk reduction. Several previous studies had already discussed the implementation of constructed wetlands as decentralized approaches for wastewater management in Leh Town [54,66–68]. However, a scale-up of the systems for the whole town and an analysis of feasible bare and agricultural land for local wastewater treatment and irrigation had not been conducted previously. The spatial analysis enabled by these tools was also useful to make city-level estimates of WEF benefits. As this was enabled through the use of openly available GIS software and land-use data, the use of these tools could be extrapolated to other locations where data availability and local capacity for data collection are limited. However, the proposed use of GISs can also have limitations regarding site-specific data availability, local lack of training to use the software and limited local capacity to update databases, as pointed out in [61]. Although there is openly available geographic data that can support decision making, there is still place-specific information that needs to be provided by the community, stakeholders and local institutions. The lack of capacity to conduct such data collection can be a limitation for further application of this method in other locations.

The analysis of the local WEF sectors and the use of spatial analysis to operationalize possible risk management approaches used in this work address some of the key issues for WSP implementation found in the literature. The need for adaptation to local contexts, support of partners external to the water supplier and demonstration of WSPs as the basis for investment [16,88] can be supported by showing the intersectoral benefits that the WEF Nexus approach has within CR-WSPs. In addition, addressing the WEF Nexus helps to create a supportive environment for climate resilience, as this requires intersectoral collaboration and policies [26]. However, assessing to which extent this applies within manageable complexity with strong stakeholder interaction at city level in this and other locations is a matter of further work. Furthermore, a reported limitation of the WEF Nexus approach is found in the lack of empirical evidence of the linkage between the application of this concept and an actual improvement in resource management and governance [89]. Due to the broadness of the WEF Nexus approach, there is a wide range of objectives, methods and conclusions obtained from its application which limits the ability to operationalize or draw overarching lessons from it [89]. An attempt to address these limitations is proposed through this work, where the WEF Nexus approach is proposed to be implemented within an already established risk management framework. As these results are, so far, purely theoretical, the actual impact of the proposed solutions needs to be evaluated after their implementation and discussion with the related stakeholders.

Although the most critical hazard for the water system in Leh Town was addressed in the proposed scenarios, other risks identified in the CR-WSP were not studied in this work.

An important hazard to consider relates to the possible increase in flashflood frequency derived from climate change in the region. For instance, a setback distance for identifying the available bare land for installing the CWs was not considered, as there is no available information as to flashflood risk-zoning maps. This is a matter of further study within the risk mitigation actions for the WEF system in Leh Town. Another important hazard not studied in this work concerns the increase in water consumption arising from the expansion of tourism and economic growth of the town. Demand management is also an important part of the principles of IWRM at a city scale and should also be addressed through other mechanisms. Examples of possible ways include the promotion of the use of traditional Ladakhi dry toilets, education campaigns on conscious water use for tourists or economic instruments such as water tariffs. Other measures for replacing fossil fuels in the drinking-water provision systems by implementing solar-powered pumps can also be considered. In addition, aquifer protection measures that will protect the quality of the source should also be considered.

This work used the urban critical infrastructure approach to analyze the interdependencies between the WEF sectors in Leh Town. Although this proved to be useful for identifying the most critical interactions, a more comprehensive assessment that evaluates other important elements of resilience and WEF security could provide a stronger basis for risk assessment and management. For instance, the preparedness, leadership, technical capacity and intersectoral cooperation within existing and planned policies are important dimensions to evaluate the resilience at urban level [4]. The integration of criteria that describe the ecological state of the water environment, human health and wellbeing, the sustainability of livelihoods and the societal stability, functions and responsibility is an additional aspect that can support a more thorough assessment of the WEF security [7]. In any case, the implementation of the WEF Nexus approach in the entire CR-WSP process needs to be supported by a stakeholder approach that not only includes the institutions responsible for water management but also those responsible for city planning, energy and food provision. Consequently, the results of this work still need to be discussed within a stakeholder approach. Furthermore, establishing roles and responsibilities is one of the key policy implementation issues identified by the Water and Sanitation Program in India [16,88]. The intersectoral approach that the WEF Nexus approach requires increases the complexity and thus a clearer definition of the functioning of intersectoral collaboration needs to be studied. Ways to managing this new complexity need to be explored particularly in developing rural areas where the capacity can be limited.

Lastly, as urban WEF issues are dynamic and socioeconomic, environmental and governance drivers transform over time [2], incorporating the transient character of the hazards on the WEF system at urban level can further improve the analysis. The scenarios considered in the present work only analyzed present states of water consumption, land use and resource management strategies. Further work including system dynamics can help to better understand sectoral interconnections and their development over time, which could provide a better insight into the WEF system in a climate change context.

## 6. Conclusions

This work exposed two possible ways for implementing the WEF Nexus approach into the CR-WSP methodology using Leh Town, India, as a case study. First, by including a critical infrastructure analysis of the WEF sectors in the system description of the CR-WSP. Second, by proposing scenarios for risk management that exploited potential WEF synergies which contributed to an enhanced WEF security in the town. On one side, the critical infrastructure analysis of the WEF sectors in the study area helped to explicitly highlight and characterize the current interdependencies as well as their implications facing the effects of a changing climate. The involvement of WEF sectors in the system description also aided the selection of suitable alternatives for risk management that considered the intersectoral dependencies and extended the benefits to sectors other than water. In this sense, addressing the WEF Nexus within the CR-WSP approach enabled the consideration

of solutions that enhanced intersectoral collaboration instead of limiting to more conventional water management approaches that solely focus on implementing strategies for a single sector. Hence, the present work serves as an example of the benefits of implementing the WEF Nexus approach in a framework for managing urban water-related risks, which can further contribute to formulating more robust climate change adaptation strategies. The use of this approach for water resource management in other cities could likewise strengthen the resilience of their WEF systems by providing alternatives for adaptation to changes in resource availability, reduction of resource use competition and promotion of intersectoral collaboration and policy coherence. Ultimately, this enhances the response capacity of urban WEF systems, contributing to the reduction of their vulnerability.

This work further exemplified how implementing a WEF Nexus strategy for water reclamation in Leh Town showed improvements in terms of risk reduction of groundwater contamination from seepage of on-site treatment systems, as well as potential for local production of reclaimed water, fertilizer and crop production. Through spatial analysis and modeling of water reclamation scenarios, it could be demonstrated that there is a potential volume of reclaimed water available for irrigation of 275 $m^3$/d, which could cover the irrigation demand for about 55 ha of agricultural land. The implementation of the scenarios could enable the recovery of up to 155 t/y of fertilizer from sludge, which could potentially replace the entirety of the demand in the district. The spatial analysis also showed that there is enough available bare and agricultural land for local implementation of the decentralized water reclamation systems. The proposed scenarios also align with the current development vision of Leh Town, in which improving sanitation coverage, implementing water reclamation, reducing the dependency of food imports, enhancing food production, implementing water-wise irrigation and organic crop production practices are key identified action points for a sustainable high-altitude town [39,85]. These results illustrate how cities can benefit from the implementation of a WEF Nexus approach within their CR-WSPs, as it can help to mitigate water-related risks while also having positive impacts on food and energy security. Implementing such a strategy at urban level should concur with the local conditions, values and goals in terms of resource use and climate change mitigation and adaptation agenda. This way, cities can envision strategies that simultaneously serve different objectives while encouraging trans-sectoral coordination and collaboration between stakeholders with competing needs, thus strengthening the resilience of their WEF systems.

The use of risk maps, created with a simplified, semi-quantitative spatial model, allowed risk visualization and showed agreement with previously performed water quality measurements. The use of risk mapping also allowed us to identify the changes in the spatial distribution of the risks when the recommended scenarios were implemented. For this purpose, the most important variables that influenced the seepage of wastewater into the groundwater of Leh Town were identified, categorized and operationalized using literature, survey data and feedback from local stakeholders. As this was supported with the use of openly available GIS software and land-use data, the use of these tools could be extrapolated to other locations where data availability and local capacity for data collection are limited. Nevertheless, this approach also requires site-specific information from the community, sectors and local institutions. A robust stakeholder involvement strategy is therefore vital for its application in other urban settings, as well as for the evaluation of its suitability for managing particular local risks.

Achieving water security is crucial to meet the international agenda for development and requires intensive intersectoral collaboration for sustainable and resilient interventions. Current risk-based approaches for enhancing water security and increasing resilience against the effects of climate change, such as CR-WSPs, can benefit from the integration of the WEF Nexus approach. Considering the WEF interdependencies, synergies and trade-offs at urban level can help to envision solutions for managing water-related risk and have important positive impacts on overall WEF security. The method for integrating the WEF Nexus approach in CR-WSPs provided through this work can serve as a base to

develop mechanisms for city planning processes that contribute to the shift from a sectoral to a trans-sectoral, resilient approach.

**Author Contributions:** In the writing of this paper, N.P.-C. contributed to the conceptualization; design of the methodology; utilization of the GIS software and mapping; formal analysis and validation; original draft preparation; and visualization of results. D.K.-G. and J.E.D. contributed to conceptualization of the research; design of the research methodology; writing, reviewing and editing the paper; and supervision of the research work. All authors have read and agreed to the published version of the manuscript.

**Funding:** This research received no external funding.

**Institutional Review Board Statement:** Not applicable.

**Informed Consent Statement:** Not applicable.

**Data Availability Statement:** Not applicable.

**Acknowledgments:** We gratefully acknowledge the outstanding support from the Ladakh Ecological Development Group (LEDeG) in the whole development of this research. Map data copyrighted: OpenStreetMap contributors and available from https://www.openstreetmap.org (accessed on 16 September 2021).

**Conflicts of Interest:** The authors declare no conflict of interest.

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
