# Peer review of "Application of the Water–Energy–Food Nexus Approach to the Climate-Resilient Water Safety Plan of Leh Town, India"

_sustainability, doi:10.3390/su131910550_

Round 1

Reviewer 1 Report

The article is well established in the literature.

The broad perspective of the authors takes into account comprehensively natural, social and economic aspects.

Logical and correct approach to the presented issues. The methodology used is correct.. Correct and convincing argumentation.

The inference based on a very good knowledge of the region.

The authors present the current situation objectively, present risks and forecast development scenarios.

The proposed solutions, such as potential WEF synergies, are of great practical importance - the article should be published and reach the decision-makers.

minor attention

Chapter 2. Case Study Site: Leh Town (Ladakh, India) should be supplemented with a map with the location of the research area.

Author Response

We thank you and gratefully acknowledge your valuable comments. We have included a map of the location of Leh Town in Chapter 2, as you suggested.

Reviewer 2 Report

This study provided the application of the water-energy-food Nexus approach to the climate-resilient water safety plan in India, however, after going through this manuscript, I feel that the paper could be improved to provide more robust and generic conclusions. Specifically the paper could be improved through the following means:

  1. It is recommended to condense the language and reduce the length of the article. Too much detail is spent in the introduction and the case.
  2. Some of the literature is missing on Line 896. Please check!
  3. Results and discussion: The author should also discuss the universality and defects of the WEF-Nexus method, In addition, please provide more practical and inductive conclusions.

Author Response

Thank you very much for your insightful comments. We have acknowledged your feedback in several sections of the manuscript, as described below:

  • We agree that it was necessary to describe the contents of chapters 1 and 2 more succinctly. We removed some paragraphs from Chapter 1 (Introduction) and replaced them with a single paragraph summarizing the essential messages. In Chapter 2 (Case study description), we deleted section 2.4. and condensed its main ideas in the description of Leh Town’s water sector. We also deleted section 2.5. and summarized it into Chapter 3 (Methodology), specifically in section 3.3., where the constructed wetland scenarios are described.
  • We have included subchapter 1.3 to state more clearly the research gap and hypothesis. This section was built by relocating some paragraphs from former subchapters (1.1. and 1.2.).
  • Thank you for pointing out the missing reference on line 896. We corrected this mistake.
  • We agree that we were lacking a discussion on the universality and limitations of the WEF-Nexus approach. This is now acknowledged in Chapter 4 (Discussion) in paragraphs 2, 3, and 4.
  • We also agree that more practical and inductive conclusions were missing in the manuscript, which are now enclosed in paragraphs 1, 2 and 3 of Chapter 5 (Conclusions). Overall, the changes implemented aimed to link the results with the broader implications of implementing the WEF-Nexus approach into CR-WSPs, which showed potential to contribute to a reduced vulnerability and increased resilience towards climate change at urban level. The changes made do not affect the original interpretation of our findings.